# SINGLE-IMAGE COHERENT RECONSTRUCTION OF OBJECTS AND HUMANS

## ABSTRACT

Existing methods for reconstruction of objects and humans from a monocular image suffer from severe mesh collisions and performance limitations for interacting occluding objects. In this paper, we introduce a method that deduces spatial configurations and achieves globally consistent 3D reconstruction for interacting objects and people captured within a single image. Our contributions encompass: 1) an optimization framework, featuring a novel collision loss, tailored to handle complex human-object and human-human interactions, ensuring spatially coherent scene reconstruction; and 2) a novel technique for robustly estimating 6 degrees of freedom (DOF) poses, particularly for heavily occluded objects, exploiting image inpainting. Notably, our proposed method operates effectively on images from real-world scenarios, without necessitating scene or object-level 3D supervision. Through both qualitative and quantitative assessments, we demonstrate the superior quality of our reconstructions, showcasing a significant reduction in collisions in scenes with multiple interacting humans and objects.

## 1 INTRODUCTION

Existing methods for human and object reconstructions are either limited to single objects and humans or give limited performance for complex images with multiple people and objects Kanazawa et al. (2018); Kolotouros et al. (2019a); Choy et al. (2016); Girdhar et al. (2016); Hassan et al. (2019); Savva et al. (2016). These methods either estimate the 3D poses of humans and objects independently or do not take into account the human-human interactions Zhang et al. (2020) and even if they do they generally follow a supervised approach Jiang et al. (2020). This leads to large collisions between the meshes with incoherent reconstructions. In this paper, we consider the full scene holistically and exploit information from the human-human and human-object interactions for spatially coherent and more complete 3D reconstruction of in-the-wild images.

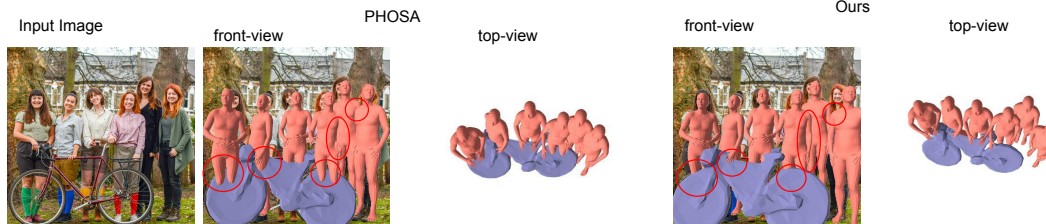

Figure 1: Comparison of the proposed method (right) reconstruction with PHOSA(middle). The proposed method gives a more coherent reconstruction with correct spatial arrangement by reasoning about human-human and human-object interaction

PHOSA Zhang et al. (2020) pioneered the field and proposed the first method that reconstructs humans interacting with objects for complex images. However, PHOSA excludes human-human interactions and gives erroneous reconstructions when objects are heavily occluded, which leads to reconstructions with incorrect depth ordering and mesh collisions. Multi-human model-free reconstruction from a single image was proposed in Mustafa et al. (2021), however, this method does not deal with interacting humans. Other methods Sun et al. (2021; 2022) for multi-person reconstructions give spatially incoherent reconstructions with severe mesh collisions because they reconstruct

each person independently. To address these challenges, in this paper, we have proposed a novel optimization-based framework for the spatially coherent reconstruction of scenes encompassing both people and objects and can deal with heavy occlusions of the objects by humans. The method first reconstructs humans Joo et al. (2021) and objects Kato et al. (2018) in the image independently. The initial poses of people in the scene are optimized to resolve any ambiguities that arise from this independent composition using a novel collision loss, depth ordering, and interaction information. To deal with heavily occluded objects, a novel 6 DOF pose estimation is proposed that uses inpainting to refine the segmentation mask of the occluded object for significantly improved pose estimation. Finally, we propose a global objective function that scores distinct 3D object layouts, orientations, collision, and shape exemplars. Gradient-based solvers are used to obtain globally optimized poses for humans and objects. Our contributions are:

- The first approach that generates a unified coherent reconstruction of a scene from a single image by effectively capturing interactions between humans and between humans and objects, all without relying on any 3D supervision.
- A novel collision loss in an optimization framework to robustly estimate 6 DOF poses of multiple people and objects in crowded images.
- An inpainting based method to improve the segmentation mask of heavily occluded objects that greatly boosts the precision of 6 DOF object position estimations.
- Qualitative and quantitative evaluation of the proposed method on complex images with multiple interacting humans and objects from the COCO-2017 dataset Lin et al. (2014) against the state-of-the-art demonstrate the effectiveness of our approach.

## 2 RELATED WORK

**3D humans from a single image:** Methods like Loper et al. (2015); Pavlakos et al. (2019); Zhou et al. (2010) use statistical body models and a large number of 3D scans to recover 3D humans from a single image. Bogo et al. (2016) use 2D poses, Tung et al. (2017) uses 2D body joint heatmaps and Kolotouros et al. (2019b) uses GraphCNN to estimate SMPL model Loper et al. (2015). However these methods only estimate 3D of single person in the scene. Methods like Zanfir et al. (2018a;b); Jiang et al. (2020); Mustafa et al. (2021) recover the 3D poses and shapes of multiple people focus on resolving ambiguities that arise due to incorrect depth ordering and collisions between people. However these methods cannot handle large occlusions. In our work, inspired from Jiang et al. (2020) we propose the collision loss and the depth ordering loss and use the 3D regression network proposed in Omran et al. (2018) to recover the 3d pose and shape of interacting humans.

**3D objects from a single image:** The earlier techniques for single-view 3D object reconstruction Lim et al. (2013); Kholgade et al. (2014); Aubry et al. (2014); Kar et al. (2015) estimate deformable shape models via optimization. Recent methods train deep networks to estimate the 3D form from an image Choy et al. (2016); Girdhar et al. (2016); Fan et al. (2017); Groueix et al. (2018), exploiting multi-view cues with 3D supervision. These methods give incorrect 3D spatial arrangement. Recently, Kulkarni et al. (2019) used GNN trained on a synthetic dataset without any humans to deduce an object's layout. In this paper we use a differentiable renderer to determine the object's initial 6 DOF pose and then fine-tune the poses by taking into account the human-object and human-human interactions without any 3D supervision.

**3D human-to-object interaction:** Savva et al. (2016) used information from the RGBD videos of individuals interacting with interiors to train a model that understands how people interact with their surroundings. Access to 3D scenes gives scene constraints that enhance the perception of 3D human poses Yamamoto & Yagishita (2000); Rosenhahn et al. (2008); Kjellström et al. (2010). Hassan et al. (2019) uses an optimization-based method to enhance 3D human posture estimates conditioned on a particular 3D scene obtained from RGBD sensors. Another recent method, Rosinol et al. (2020), creates a 3D scene graph of people and objects for indoor data. Chen et al. (2019) represents the optimal configuration of the 3D scene, in the form of a parse graph that encodes the object, human pose, and scene layout from a single image. In our work, we overcome the limitations of existing methods by handling not only on human-object interactions but also capturing human-human interactions and also propose a method that deals with major occlusions that result in significantly improved scene reconstruction.

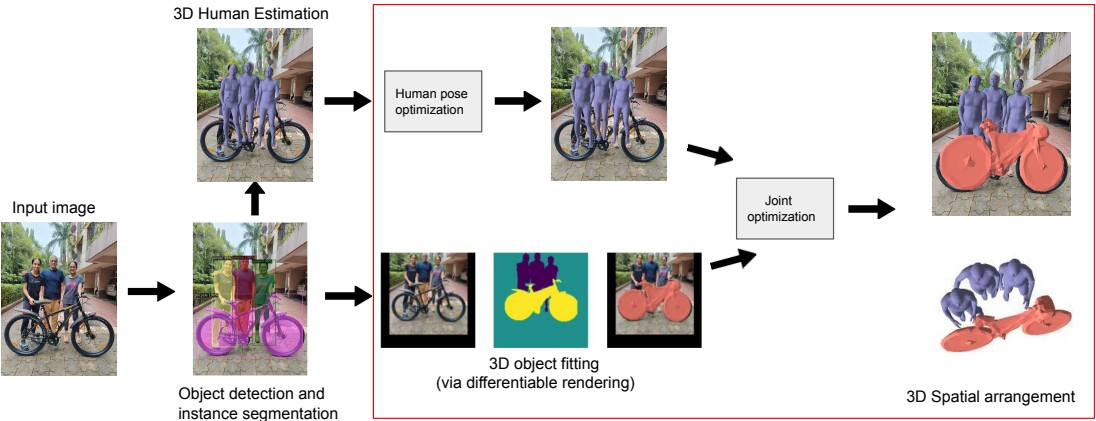

Figure 2: Overview of the proposed method to generate spatially coherent reconstruction from a single image. The steps in red box are novel. The reconstruction before human pose optimization exhibits notable mesh collisions. After human pose optimization, reduced mesh collisions are seen while maintaining relative coherence between humans.

## 3   METHODOLOGY

The proposed method takes a single RGB image as input and gives a spatially coherent reconstruction of interacting humans and objects in the scene, an overview is shown in Figure 2. We exploit human-human and human-object interactions to spatially arrange all objects in a common 3D coordinate system. First, objects and humans are detected followed by SMPL based per person reconstruction (Sec. 3.1), which gives incorrect spatial reconstructions and collisions. The human 3D locations/poses are translated into world coordinates and refined through a human-human spatial arrangement optimization using a novel collision loss (Sec. 3.2). To correctly estimate the 3D object pose (6-DoF translation and orientation) a differentiable renderer is used that fits 3D mesh object models to the predicted 2D segmentation masks Kirillov et al. (2020). We correct the occluded object mask using image inpainting (in Sec. 3.3) unlike PHOSA Zhang et al. (2020) which uses an occluded object mask. Lastly, we perform joint optimization that takes into account both human-human and human-object interactions for a globally consistent output. Our framework produces plausible reconstructions that capture realistic human-human and human-object interactions.

### 3.1   ESTIMATING 3D HUMANS

Using Joo et al. (2021), we estimate the 3D shape and pose parameters of SMPL Loper et al. (2015) given a bounding box for a human Mask (2017). The 3D human is parameterized by pose $\theta \in \mathbb{R}^{72}$, shape $\beta \in \mathbb{R}^{10}$, and a weak-perspective camera $\gamma = [\sigma, t_x, t_y] \in \mathbb{R}^3$. To position the humans in the 3D space, $\gamma$ is converted to the perspective camera projection by assuming a fixed focal length f for all images, and the distance of the person is determined by the reciprocal of the camera scale parameter $\sigma$. Thus, the 3D vertices of the SMPL model for the $i^{th}$ human are represented as: $M_i = J(\theta_i, \beta_i) + [t_x, t_y, \frac{f}{\sigma^i}]$, where $J$ is the differentiable SMPL mapping from pose and shape to a human mesh and $t_h^i = [t_x, t_y, \frac{f}{\sigma^i}]$ is the translation of $i^{th}$ human. The person's height and size are regulated by the SMPL shape parameter $\beta$. We define scale parameter($s^i$) for each human similar to PHOSA and the final vertices are given by $\bar{M}_i = s^i M_i$.

### 3.2   HUMAN POSE OPTIMISATION

Independently analyzing human 3D poses results in inconsistent 3D scene configurations. Human-human interactions can offer useful information to determine the relative spatial arrangement and not considering this leads to ambiguities like mesh penetration and incorrect depth ordering. We present an optimization framework that incorporates human-human interactions. We first identify interacting humans in the image and then optimize the pose through an objective function to correctly adjust their spatial arrangements.

**Identifying interacting humans -**  Our hypothesis posits that human interactions are contingent upon physical proximity in world coordinates. Hence we find the interacting humans by identifying

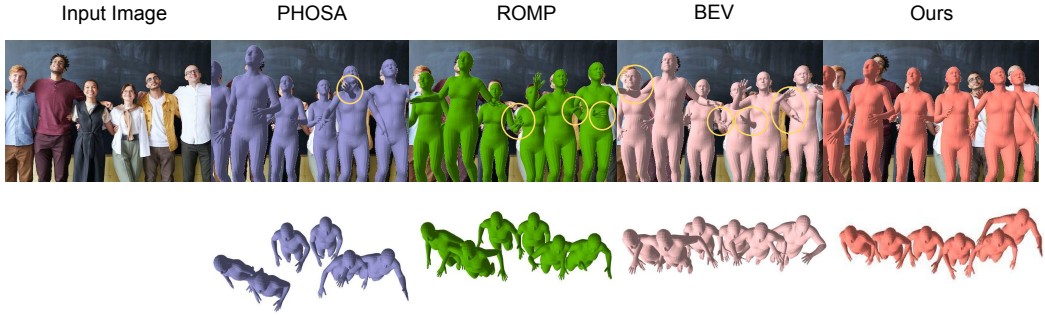

| Input Image | PHOSA | ROMP | BEV | Ours |

Figure 3: The proposed approach produces spatially coherent reconstructions with a significant reduction in mesh collisions compared to PHOSA Zhang et al. (2020), ROMPSun et al. (2021), and BEVSun et al. (2022). Significant collision focal points are visually emphasized through discernible encirclements within the presented image.

the overlap of 3D bounding boxes(More details regarding bounding box overlap criteria can be found in the appendix C).

**Objective function to optimize 3D spatial arrangement -** Our objective includes collision, interaction, and depth ordering loss terms to constrain the pose for interacting humans:

$$L_{HHI-Loss} = \lambda_1 L_{H-collision} + \lambda_2 L_{H-depth} + \lambda_3 L_{H-interaction} \tag{1}$$

We optimize (1) using a gradient-based optimizer Kingma & Ba (2014) w.r.t. translation $\mathbf{t}^i \in \mathbb{R}^3$ and scale parameter $s^i$ for the $i^{th}$ human instance. The human translations are initialized from Sec 3.1. The terms in the objective function are defined below:

**Collision Loss ($L_{H-collision}$) -** To overcome the problem of mesh collisions, as seen in existing methods in Figure 3.2, we introduce a novel collision loss $L_{H-collision}$ that penalizes interpenetrations in the reconstructed people. Let $\phi$ be a modified Signed Distance Field (SDF) for the scene that is defined as follows: $\phi(x, y, z) = -min(SDF(x, y, z), 0)$ where $\phi$ is positive for points inside the human and is proportional to the distance from the surface, and is 0 outside of the human. Typically $\phi$ is defined on a voxel grid of dimensions $N_p * N_p * N_p$. While it's definitely possible to generate a single voxelized representation for the entire scene, we often find ourselves requiring an extensive fine-grained voxel grid. Depending on the scene's extent, this can pose processing challenges due to memory and computational limitations. To overcome this a separate $\phi_i$ function is computed for each person by calculating a tight box around the person and voxelizing it instead of the whole scene to reduce computational complexity Jiang et al. (2020). The collision penalty of person $j$ for colliding with person $i$ is defined as follows: $P_{ij} = \sum_{v \in M_j} \tilde{\phi}_i(v)$, where $\tilde{\phi}_i(v)$ samples the $\phi_i$ value for each 3D vertex $v$ in a differentiable way from the 3D grid using trilinear interpolation. If there is a collision between person $i$ and a person $j$, $P_{ij}$ will be a positive value and decreases as the separation between them increases. If there is no overlap between person $i$ and $j$, $P_{ij}$ will be zero. Let the translation of person $i$ and person $j$ be $T_i$ and $T_j$ respectively. Then the collision loss between them is defined as:

$$L_{ij} = \begin{cases} \frac{P_{ij}}{\exp(||T_i - T_j + \delta||_2)} & T_i = T_j \\ \frac{P_{ij}}{\exp(||T_i - T_j||_2)} & T_i \neq T_j \end{cases} \tag{2}$$

When the translation values are the same (in case of maximum overlap) we use an extra term $\delta$ ($0 < \delta < 1$) to ensure non-zero gradients are not very large to avoid any instabilities during optimization. The final collision loss for a scene with N people is defined as follows:

$$L_{H-collision} = \sum_{j=1}^{N} \left( \sum_{i=1 i \neq j}^{N} L_{ij} \right) \tag{3}$$

**Interaction Loss ($L_{H-interaction}$) -** This is an instance-level to pull the interacting people close together, similar to Zhang et al. (2020): $L_{H-interaction} = \sum_{h_i, h_j \in H} \mu(h_i, h_j) ||C(h_i) - C(h_j)||_2$, where $\mu(h_i, h_j)$ identifies whether human $h_i$ and $h_j$ are interacting according to the 3D bounding box overlap criteria. $C(h_i)$ and $C(h_j)$ give the centroid for human $i$ and human $j$ respectively.

**Depth-Ordering Loss ($L_{H-depth}$) -** To achieve more accurate depth ordering we refer to the depth loss, as in Jiang et al. (2020). The loss is defined as: $L_{depth} = \sum_{p \in S} \log(1 + \exp(D_{y(p)}(p) - $

$D_{\bar{y}(p)}(p)))$, where $S = \{p \in I : y(p) > 0, \bar{y}(p) > 0, y(p) \neq \bar{y}(p)\}$ is the pixels in the image I with incorrect depth ordering in the ground truth segmentation, the person index at pixel position $p$ is represented by $y(p)$, and the predicted person index in the rendered 3D meshes is $\bar{y}(p)$ and $y(p) \neq \bar{y}(p)$. $D_{y(p)}(p)$ and $D_{\bar{y}(p)}(p)$ represent the pixel depths.

### 3.3 3D Object Pose Estimation

After estimating the shape and pose of humans, the next step is to estimate the same for the objects. To estimate the 3D location $t \in \mathbb{R}^3$ and 3D orientation $R \in SO(3)$ of the objects. For each object category, exemplar mesh models are pre-selected. The mesh models are sourced from Fre; Kundu et al. (2018). The vertices of $j^{th}$ object are: $V_o^j = s^j(R^j O(c^j, k^j) + t^j)$, where $c^j$ is the object category from MaskRCNN Mask (2017), and $O(c^j, k^j)$ determines the $k^j - th$ exemplar mesh for category $c^j$. The optimization framework chooses the exemplar that minimizes re-projection error to determine $k^j$ automatically and $s^j$ is the scale parameter for $j^{th}$ object.

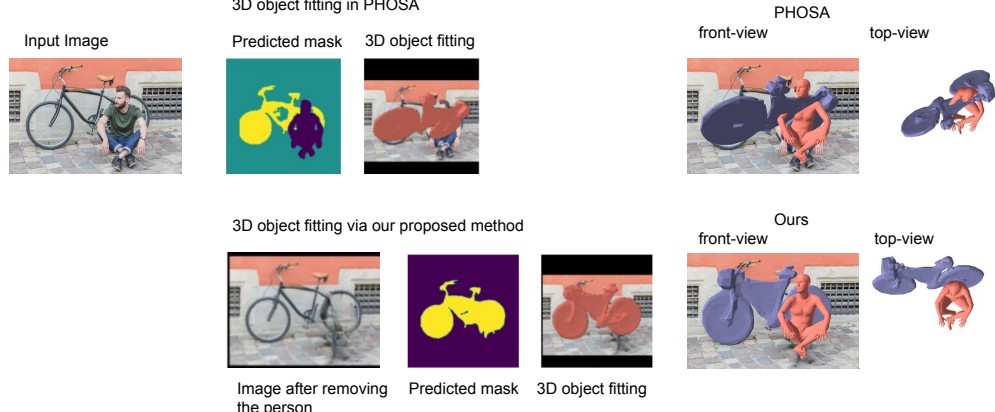

Figure 4: Comparison of the segmentation masks and reconstruction with PHOSA. The segmentation mask of the bicycle is occluded resulting in erroneous reconstruction in PHOSA. The proposed method uses image inpainting to remove the occlusion to generate a better segmentation mask, which leads to a more complete reconstruction.

Our first objective is to estimate the 6 DOF pose of each object independently. It is difficult to estimate 3D object pose in the wild as there are: (1) no parametric 3D models for objects; (2) no images of objects in the wild with 2D/3D pose annotations; and (3) occlusions in cluttered scenes with humans. We address these challenges by proposing an optimization-based approach that uses a differentiable renderer Kato et al. (2018) to fit the 3D object to instance masks from Kirillov et al. (2020) in a manner that is robust to minor/major occlusions.

As defined in Zhang et al. (2020) we calculate a pixel-wise L2 loss over rendered silhouettes S versus predicted masks M but the quality of the predicted mask M is impacted by occlusions as seen in Zhang et al. (2020), which results in a poorly estimated 6 DOF pose. To address problems due to occlusions, we propose a novel method that improves the masks as shown in Figure 3.3.

Given an image $I$, a total number of objects $N$, and bounding boxes for rigid $B_r$ and non-rigid $B_{nr}$ objects, along with their masks - $M_r$ for rigid and $M_{nr}$ for non-rigid objects. Each $i^{th}$ object can be occluded by maximum $N - 1$ objects. To identify occluding objects we calculate the Intersection over Union(IOU) between all pairs of bounding boxes. Objects with $IOU > 0.3$ (Our selection of this threshold stems from our empirical observations, wherein we found that objects with $IOU > 0.3$ led to noticeable improvements in reconstruction quality. Conversely, when IOU was less than 0.3, the reconstruction results obtained using our method closely resembled those produced by PHOSA Zhang et al. (2020), more details in appendix D) are occluding objects $M$ for each object. Occluding objects can be removed in numerous ways, for e.g remove only one object at a time. The total possible combinations, in this case, are $\binom{M}{1}$, or you remove a pair of objects at a time and the total possible combinations, in this case, are $\binom{M}{2}$ and so on. The total number of all possible combinations can be described as $\binom{M}{0} + \binom{M}{1} + \binom{M}{2} + ..... \binom{M}{M} = 2^M$. To remove $j$

occluding objects where $j \leq M$ we need a single mask $M_{occ-mask}$ that is a combination of the $j$ masks, so $M_{occ-mask} = M_1 + M_2 + .... + M_j$. Now we use the image-inpainting approach proposed by Nazeri et al. (2019) to remove the occluding objects. We pass the current image $I$ and the mask $M_{occ-mask}$ to get a new image without occlusions and use this image to get the new segmentation masks and bounding boxes:

$$I_{new} = EC(I, M_{occ-mask})$$
$$B_r^{new}, B_{nr}^{new}, M_r^{new}, M_{nr}^{new} = OD(I_{new}) \tag{4}$$

where $EC$ is the image inpainting algorithm and $OD$ is the object detection algorithm. Sometimes, the $i^{th}$ object in $I$ may not correspond to the same object in $I_{new}$. Let's say the index of the $i^{th}$ object in $I_{new}$ be $k$. We iterate over the list of new bounding boxes and calculate the $IOU$ of these boxes with $B_r[i]$ and, $k$ corresponds to the index of the bounding box for which $IOU$ is closest to 1. We use the mask $M_r^{new}[k]$ to determine object pose. Estimating a reliable pose also depends heavily on the boundary details. To incorporate this we augment the L2 mask loss with a modified version of the symmetric chamfer loss Gavrila (2000). Given a no-occlusion indicator $\eta$ (0 if pixel only corresponds to a mask of a different instance, else 1), the loss is:

$$L_{occ-sil} = \sum (\eta \circ S - M_r^{new}[k])^2 + \sum_{p \in E(\eta \circ S)} \min_{\bar{p} \in E(M)} ||p - \bar{p_2}|| \tag{5}$$

We generate masks $M_{occ-mask}$ for different values of $j$ and a 3D pose corresponding to that mask is chosen that results in a minimum value of $L_{occ-sil}$. The edge map of mask M is computed by E(M). To estimate the 3D object pose, we minimize the occlusion-aware silhouette loss:

$$(R^j, t^j)^* = \underset{R,t}{argmin}(L_{occ-sil}(\Pi_{sil}(V_o^j), M_r^{new}[k])) \tag{6}$$

where $\Pi_{sil}$ is the silhouette rendering of a 3D mesh model via a perspective camera with a fixed focal length f (Sec 3.1) and $M_j$ is a 2D instance mask for the $j^{th}$ object. Instance masks are computed by PointRend Kirillov et al. (2020).

### 3.4 JOINT OPTIMIZATION

The joint optimization refines both the human and object poses estimated above, exploiting both human-human and human-object interactions through joint loss functions. Estimating 3D poses of people and objects in isolation from one another leads to inconsistent 3D scene reconstruction. Interactions between people and objects provide crucial clues for correct 3D spatial arrangement, which is done by identifying interacting objects and humans and proposing an objective function for refining human/object poses.

**Identifying human-object interaction**. Our hypothesis posits that human-object interactions are contingent upon physical proximity in world coordinates. We use 3D bounding box overlap between the human and object to determine whether the object is interacting with a person. (More details regarding bounding box overlap criteria can be found in the appendix C

**Objective function to optimize 3D spatial arrangements**. We define a joint loss function that takes into account both human-human and human-object interactions. It is crucial to include both of them because if you simply optimize with regard to human-object interactions, it may result in erroneous relative positions between interacting people even if it would enhance the relative spatial arrangement between the interacting humans and objects.

$$L_{joint-loss} = L_{HOI-Loss} + L_{HHI-Loss} \tag{7}$$

where $L_{HHI-Loss}$ is same as Eq. 1 and

$$L_{HOI-Loss} = \lambda_1 L_{HO-collision} + \lambda_2 L_{HO-depth} + \lambda_3 L_{HO-interaction} + \lambda_4 L_{occ-sil} \tag{8}$$

Depth-Ordering Loss ($L_{HO-depth}$) is same as Section 3.2. We optimize (8) using a gradient-based optimizer Kingma & Ba (2014) w.r.t. translation $t^i \in \mathbb{R}^3$ and intrinsic scale $s^i \in \mathbb{R}$ for the $i^{th}$ human and, rotation $R^j \in SO(3)$, translation $t^j \in \mathbb{R}^3$ and $s^j \in \mathbb{R}$ for the $j^{th}$ object instance jointly. The object poses are initialized from Sec. 3.3. $L_{occ-sil}$ is the same as (5) except without the chamfer loss which didn't help during joint optimization.

**Interaction loss** ($L_{HO-interaction}$): This loss handles both coarse and fine interaction between humans and objects as in Zhang et al. (2020), defined as: $L_{HO-interaction} = L_{coarse-inter} + L_{fine-inter}$.

The coarse interaction loss is: $L_{coarse-inter} = \sum_{h \in H, o \in O} \mu(h,o)||C(h) - C(o)||_2$, where $\mu(h,o)$ identifies whether human $h$ and object $o$ are interacting according to the 3D bounding box overlap criteria. $C(h)$ and $C(o)$ give the centroid for human and the object respectively. To handle human interactions, the fine interaction loss is defined as:
$L_{fine-inter} = \sum_{h \in H, o \in O}(\sum_{P_h, P_o \in P(h,o)} \mu(P_h, P_o)||C(P_h) - C(P_o)||_2)$, where $P_h$ and $P_o$ are the regions of interaction between the humans and the object, respectively. $\mu(P_h, P_o)$ is the overlap of the 3D bounding box between the interacting objects, recomputed at each iteration.

**Collision Loss** ($L_{HO-collision}$) - The formulation of this loss is similar to the collision loss defined in Section 3.2. The difference is that here we take into account the mesh collision between interacting humans and objects in contrast to interacting humans. Let $N_h$ represent the total number of humans and $N_o$ total number of objects, then the Loss function is defined as:
$L_{HO-collision} = \sum_{j=1}^{N_o}\left(\sum_{i=1}^{N_h} L_{h_i o_j} + L_{o_j h_i}\right)$, where $h_i$ represents the $i^{th}$ human and $o_j$ represents the $j^{th}$ object.

## 4 RESULTS AND EVALUATION

We perform both quantitative and qualitative assessments of the performance of our technique on the COCO-2017 Lin et al. (2014) dataset on images that include interactions of humans and objects against state-of-the-art methods PHOSA Zhang et al. (2020), ROMPSun et al. (2021), and BEVSun et al. (2022).

### 4.1 QUALITATIVE AND QUANTITATIVE ANALYSIS

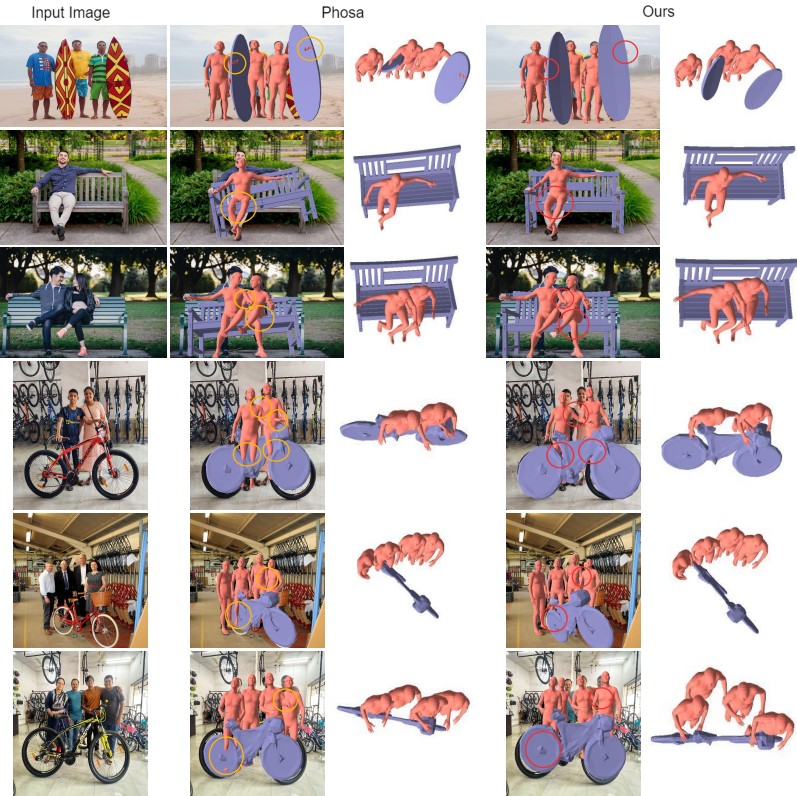

Figure 5: Qualitative comparison on test images from COCO 2017 against PHOSA Zhang et al. (2020) with human-object interactions. Our method gives better spatial reconstruction while substantially reducing collisions(the golden circles delineate regions characterized by noteworthy mesh collisions, while the red circles delineate areas showcasing enhancements in reconstructions). More qualitative results are shown in appendix E

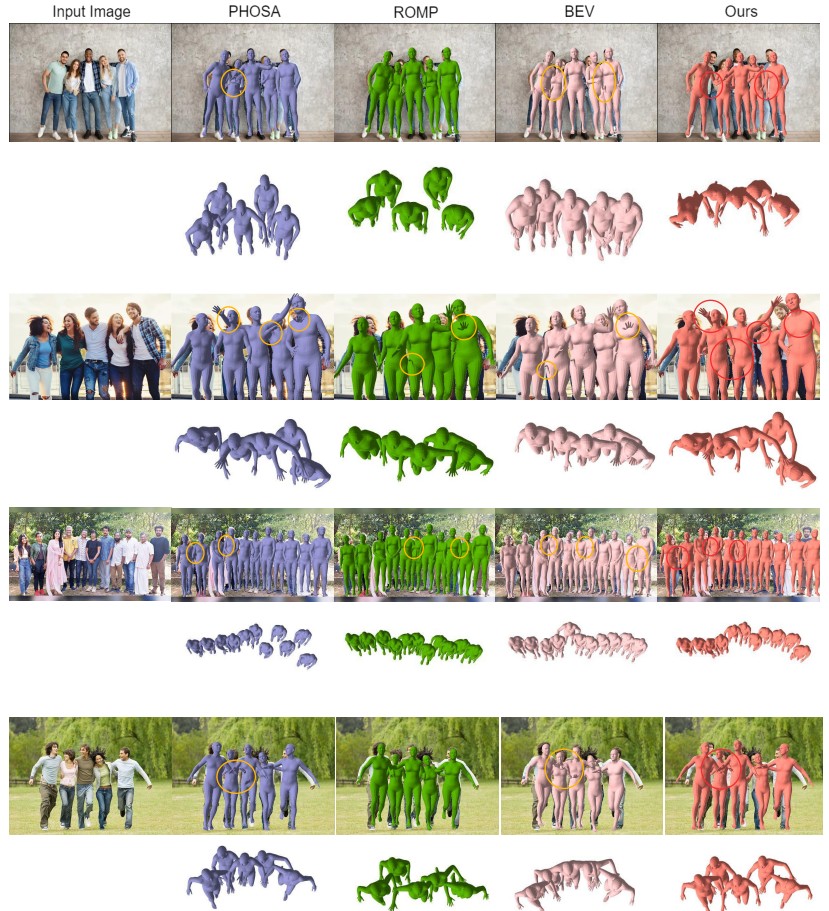

Figure 6: Qualitative results of proposed method on test images from COCO 2017 compared to PHOSA, ROMP, and BEV for human-human interactions. Our method gives more realistic and coherent reconstructions for images with multiple humans.

Figures 5 and 6 shows a **qualitative** comparison with state-of-the-art methods. PHOSA performs human-object reconstructions; and ROMP and BEV only reconstruct humans. As seen our approach yields markedly improved reconstruction quality by effectively mitigating ambiguities arising from mesh collisions and occlusions.

For **quantitative** evaluation, we employ a forced-choice assessment approach similar to PHOSAZhang et al. (2020) on COCO-2017 Lin et al. (2014) images since there are no 3D ground truth annotations for people and objects in images in the wild. From the COCO-2017 test set, we randomly selected a sample of images and performed reconstruction on each image. We compare our method with PHOSA, ROMP, and BEV by reconstructing the scenes and comparing the degree of mesh collisions for human-human $E_{H-col}$ and human-object $E_{HO-col}$ and incorrect depth ordering for human-human $E_{H-depth}$ and human-object $E_{HO-depth}$ interactions that results from each method. This is averaged across all images to estimate values in Table 1. Our approach outperforms the state-of-the-art techniques for both multi-human and multi-human-object reconstruction, as well as results in a more coherent and realistic reconstruction with significantly fewer ambiguities.

| Methods | $E_{H-col}$ | $E_{H-depth}$ | $E_{HO-col}$ | $E_{HO-depth}$ |
|---------|-------------|---------------|--------------|----------------|
| PHOSA | 196.42 | 241.68 | 257.21 | 140.84 |
| ROMP | 180.51 | 210.92 | - | - |
| BEV | 106.25 | 153.17 | - | - |
| Ours | **16.46** | **54.37** | **26.65** | **73.77** |

Table 1: Quantitative evaluation with PHOSA Zhang et al. (2020), ROMPSun et al. (2021), and BEVSun et al. (2022). BEV and ROMP only reconstruct humans. Equations of each evaluation parameter are given in the appendix.

| Ours vs. | PHOSA | ROMP | BEV |
|----------|-------|------|-----|
|  | 88% | 80% | 74% |

Table 2: User study that gives the average percentage of images for which our method performs better on COCO-2017. $50\%$ implies equal performance.

We also perform a subjective study similar to Zhang et al. (2020), where we show the reconstructions for each image from PHOSA, ROMP, BEV, and our proposed method in a random order to the users and the users mark whether our result looks better than, equal to, or worse than the other methods. We compute the average percentage of images for which our method performs better in Table 2. Overall, our method outperforms all the other methods.

| Ours vs. | No $L_{collision}$ | No $L_{depth}$ | No $L_{interaction}$ | No $L_{occ-sil}$ |
|----------|--------------------|-----------------|----------------------|-------------------|
|  | 83% | 62% | 73% | 77% |

Table 3: In the ablation study we drop loss terms from our proposed method. The higher the percentage, the more the effect of the loss term. No $L_{collision}$ implies the exclusion of both $L_{H-collision}$ and $L_{HO-collision}$. No $L_{depth}$ involves omitting $L_{H-depth}$ and $L_{HO-depth}$. No $L_{interaction}$ means we omitted the $L_{H-interaction}$ and $L_{HO-interaction}$, and lastly No $L_{occ-sil}$ corresponds to dropping the loss term defined in eq. 5

### 4.2 ABLATION STUDY

An ablative study was conducted to assess the significance of the loss terms in Table 3. The identical forced-choice test similar to PHOSAZhang et al. (2020) is conducted for the complete proposed methodology (Equation 7), by omitting loss terms from the proposed method and measuring the performance. Our findings indicate that the exclusion of the collision and occlusion-aware silhouette loss has the most notable effect, with the interaction loss following closely behind. The collision loss prevents mesh intersection and the silhouette loss guarantees that the object poses remain consistent with their respective masks.

## 5 DISCUSSION

Existing methods to reconstruct humans/objects from a single image, give incoherent reconstructions with mesh penetrations and fail for complex human-human and human-object interactions. In this paper, we perform holistic 3D scene perception by exploiting the information from both human-human and human-object interactions in an optimization framework for the first time. The optimization makes use of several novel constraints to provide a full scene that is globally consistent, and reduces collisions by $\approx 80\%$, and improves spatial arrangement by $\approx 70\%$ (Table 1) over state-of-the-art methods. The proposed human optimization framework resolves ambiguities between reconstructed people, and the proposed human-object optimization framework addresses ambiguities between humans and objects. We further introduce a method that significantly improves the pose estimation of heavily occluded objects. We demonstrate via our qualitative and quantitative evaluations that the proposed method outperforms the state-of-the-art methods and produces spatially coherent reconstructions with noticeably less ambiguity.

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

## A    IMPLEMENTATION DETAILS

The Human-Human Interaction loss - $L_{HHI-Loss}$ is optimized using ADAM Kingma & Ba (2014) with learning rate $2e-3$ for 100 iterations. The trainable parameters are translation $t^i \in \mathbb{R}^3$ and intrinsic scale $s^i \in \mathbb{R}$ for the $i^{th}$ human. We initialized the optimization with the human poses estimated using Sec 3.1(in the paper).

Using the 6-DoF rotation representation described in Zhou et al. (2019), we encode rotations for object poses. ADAM optimizerKingma & Ba (2014) is used to optimize occlusion-aware silhouette loss with a learning rate of $2e-3$ for 200 iterations and edge maps $E(M)$ are computed using $MaxPool(M)$ - $M$ with a filter size of 7.

We optimize the joint loss - $L_{joint-loss}$ using ADAM Kingma & Ba (2014) with a learning rate $3e-4$ for 500 iterations. The trainable parameters are translation $t^i \in \mathbb{R}^3$ and intrinsic scale $s^i \in \mathbb{R}$ for the $i^{th}$ human and, rotation $R^j \in SO(3)$, translation $t^j \in \mathbb{R}^3$ and $s^j \in \mathbb{R}$ for the $j^{th}$ object instance jointly. The loss weights $\lambda_i$ are tuned qualitatively on the COCO-2017 val set.We initialized the optimization with the human poses estimated in Sec 3.2 (in the paper) and the best object pose estimated in Sec.3.3 (in the paper) per object instance.

## B    EVALUATION EQUATIONS

We compare our method with PHOSA, ROMP, and BEV by reconstructing the scenes and comparing the degree of mesh collisions for human-human $E_{H-col}$ and human-object $E_{HO-col}$ and incorrect depth ordering for human-human $E_{H-depth}$ and human-object $E_{HO-depth}$ interactions that results from each method. We randomly sample the total $T$ number of images from the COCO2017 test set and for each image, reconstruction is performed via all methods to calculate $E_{H-col}$, $E_{HO-col}$, $E_{H-depth}$ and $E_{HO-depth}$. Finally, we determine the average value for all these for the different methods. A lower score implies better reconstruction. The equations used to calculate these are as follows, which indicate the average value of mesh collision loss among human-human reconstructions and humans-object reconstructions:

$$E_{HO-col} = \frac{1}{T} * \sum_{k=1}^{T} \Big( \sum_{j=1}^{N_o^k} \Big( \sum_{i=1}^{N_h^k} L_{h_i o_j}^k + L_{o_j h_i}^k \Big) \Big) \tag{9}$$

where, $L_{h_i o_j}^k$ and $L_{o_j h_i}^k$ are defined in the manuscript under Human-Object collision loss Sec 3.4 (in the paper).

where, $L_{ij}^k$ is defined in the manuscript under human-human collision loss Sec 3.2(in the paper)

$$E_{H-col} = \frac{1}{T} * \sum_{k=1}^{T} \Big( \sum_{j=1}^{N^k} \Big( \sum_{i=1 i \neq j}^{N^k} L_{ij}^k \Big) \Big) \tag{10}$$

The other evaluation parameters indicate the average value of depth disparity for human-human and human-object reconstruction across all photos.

$$E_{HO-depth} = \frac{1}{T} * \sum_{k=1}^{T} \Big( \sum_{p \in S_h U S_o} \log(1 + \exp(D_{y(p)}^k(p) - D_{\bar{y}(p)}^k(p))) \Big) \tag{11}$$

$$E_{H-depth} = \frac{1}{T} * \sum_{k=1}^{T} \Big( \sum_{p \in S_h} \log(1 + \exp(D_{y(p)}^k(p) - D_{\bar{y}(p)}^k(p))) \Big) \tag{12}$$

## C    BOUNDING BOX OVERLAP CRITERIA

Given two objects $i$ and $j$, we first determine a tight bounding box around these objects. Let us call them $box_i$ and $box_j$ respectively. One important thing to note here is that boxes are axis-aligned. Let's say that the corners of $box_i$ are $l_i = (x_1, y_1, z_1)$ and $r_i = (X_1, Y_1, Z_1)$ where $X_1 > x_1$, $Y_1 > y_1$ and $Z_1 > z_1$. Similarly, we can represent corners of $box_j$ as $l_j = (x_2, y_2, z_2)$ and $r_j = (X_2, Y_2, Z_2)$ where $X_2 > x_2$, $Y_2 > y_2$ and $Z_2 > z_2$. The way to check for an overlap is to compare the intervals $[x_1, X_1]$ and $[x_2, X_2]$, and if they don't overlap, there's no intersection. Do the same for the y intervals, and the z intervals.

```
def CheckOverlap(l_i, r_i, l_j, r_j):
    x_1, y_1, z_1 = l_i
    X_1, Y_1, Z_1 = r_i
    x_2, y_2, z_2 = l_j
    X_2, Y_2, Z_2 = r_j
    if x_1 > X_2 or x_2 > X_1:
        return false
    if y_1 > Y_2 or y_2 > Y_1:
        return false
    if z_1 > Z_2 or z_2 > Z_1:
        return false
    return true
```

## D IOU THRESHOLDING

In this paper, one of our contributions is a novel approach aimed at enhancing the segmentation mask of occluded objects. Our methodology leverages image in-painting for object removal, incorporating an Intersection over Union (IOU) threshold set at $> 0.3$. It is worth noting that this threshold is flexible, allowing for experimentation with various values. Our choice of this threshold is rooted in empirical observations, where we noted that objects with an IOU exceeding 0.3 resulted in noticeable enhancements in reconstruction quality. Conversely, when the IOU was below 0.3, the reconstruction results obtained by PHOSA closely resembled those produced by using our method. To illustrate this distinction, we provide several examples showcasing reconstruction cases where the IOU falls below and exceeds the 0.3 threshold.

IOU = 0.55

3D Object fitting in PHOSA

Input Image            Predicted mask    3D Object Fitting                    PHOSA

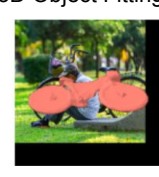 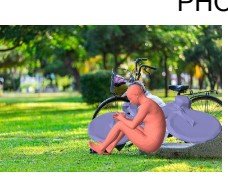 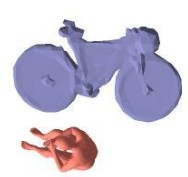

3D object fitting via our proposed method                    Ours

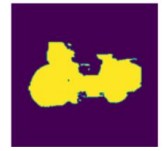 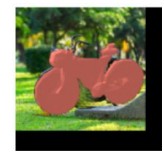 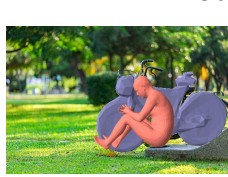 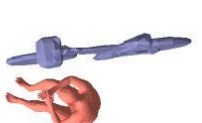

Predicted mask    3D Object Fitting

IOU = 0.32

3D Object fitting in PHOSA

Input Image            Predicted mask    3D Object Fitting                    PHOSA

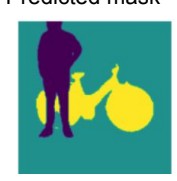 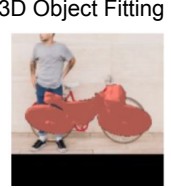 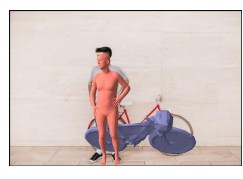 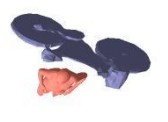

3D object fitting via our proposed method                    Ours

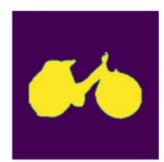 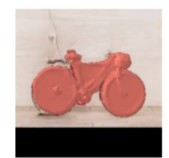 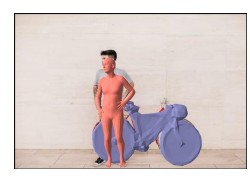 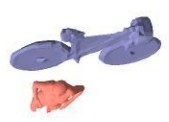

Predicted mask    3D Object Fitting

IOU = 0.21

Input Image

3D Object fitting in PHOSA

Predicted mask    3D Object Fitting

PHOSA

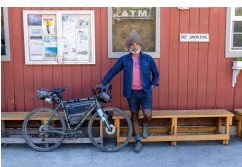
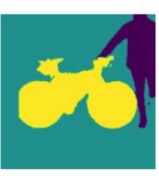
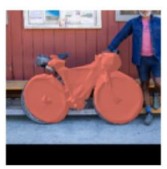
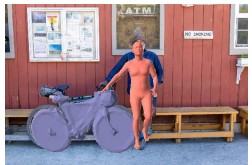
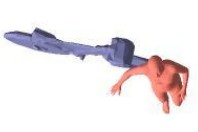

3D object fitting via our proposed method

Ours

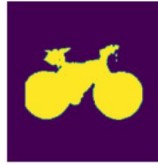
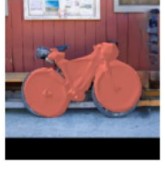
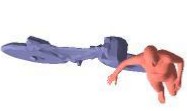

Predicted mask    3D Object Fitting

IOU = 0.16

Input Image

3D Object fitting in PHOSA

Predicted mask    3D Object Fitting

PHOSA

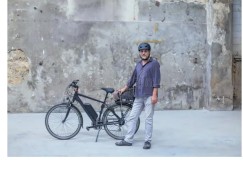
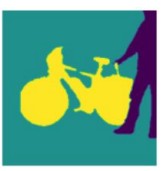
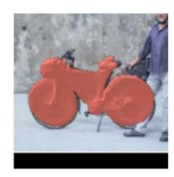
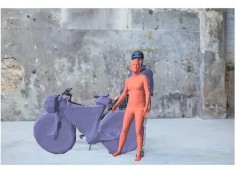
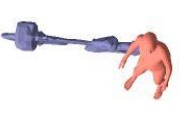

3D object fitting via our proposed method

Ours

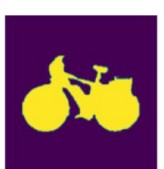
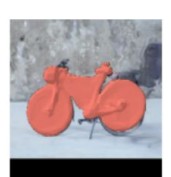
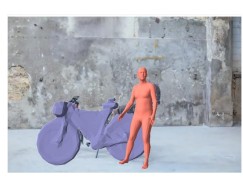
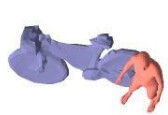

Predicted mask    3D Object Fitting

IOU = 0.45

Input Image

3D Object fitting in PHOSA

Predicted mask    3D Object Fitting

PHOSA

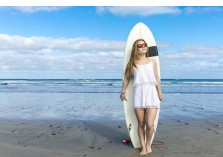 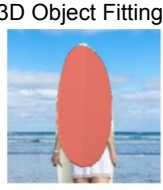 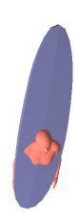

3D object fitting via our proposed method

Ours

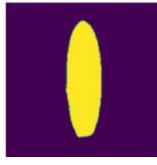 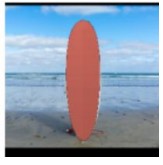 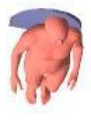

Predicted mask    3D Object Fitting

IOU = 0.37

Input Image

3D Object fitting in PHOSA

Predicted mask    3D Object Fitting

PHOSA

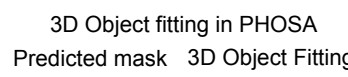

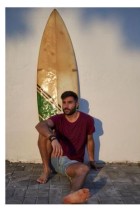 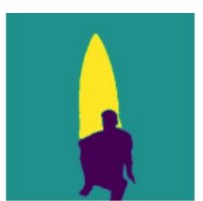 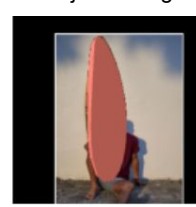 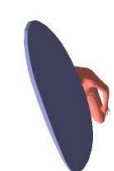

3D object fitting via our proposed method

Ours

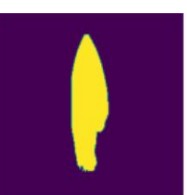 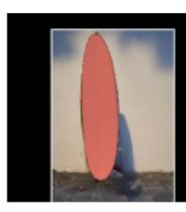 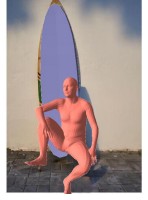 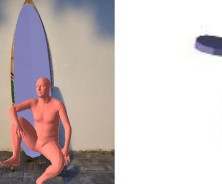

Predicted mask    3D Object Fitting

IOU = 0.27

3D Object fitting in PHOSA

Input Image    Predicted mask    3D Object Fitting                    PHOSA

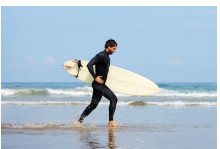 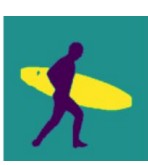 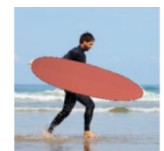          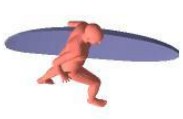

3D object fitting via our proposed method                    Ours

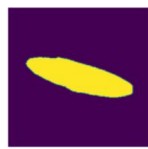 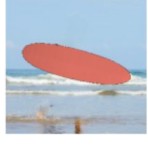                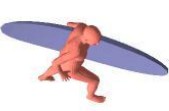

Predicted mask    3D Object Fitting

IOU = 0.18

3D Object fitting in PHOSA

Input Image    Predicted mask    3D Object Fitting                    PHOSA

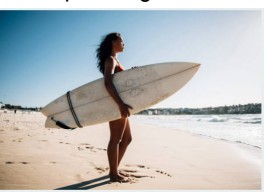 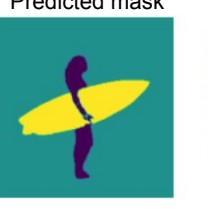 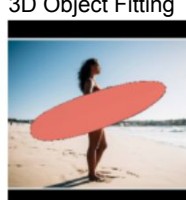    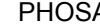 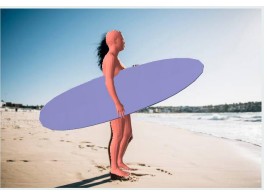 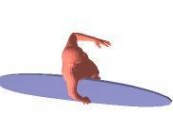

3D object fitting via our proposed method                    Ours

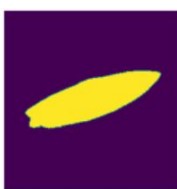 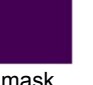 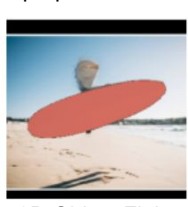    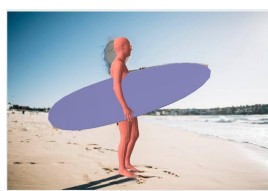 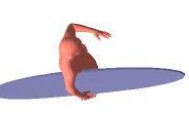

Predicted mask    3D Object Fitting

IOU = 0.71

Input Image

3D Object fitting in PHOSA

Predicted mask   3D Object Fitting

PHOSA

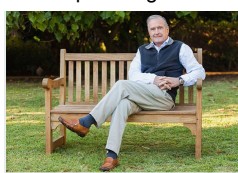
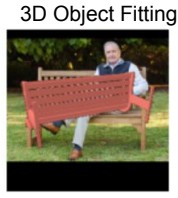
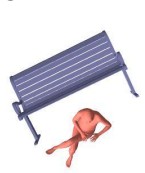

3D object fitting via our proposed method

Ours

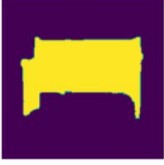
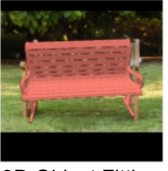
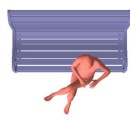

Predicted mask   3D Object Fitting

IOU = 0.28

Input Image

3D Object fitting in PHOSA

Predicted mask   3D Object Fitting

PHOSA

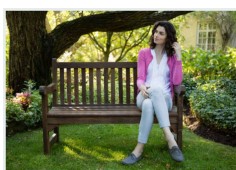
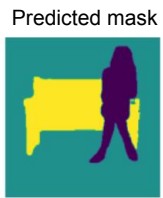
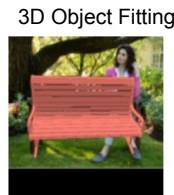
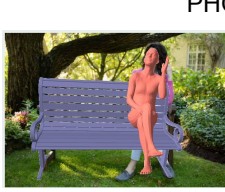
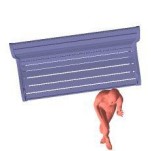

3D object fitting via our proposed method

Ours

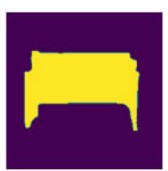
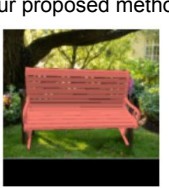
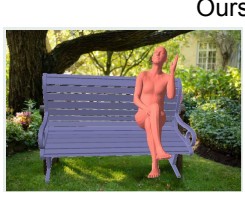
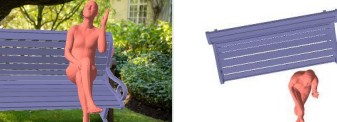

Predicted mask   3D Object Fitting

## E    MORE QUALITATIVE RESULTS

Like existing papers we have shown results on the COCO 2017 dataset. Here we show more results on challenging Youtube and Google images.

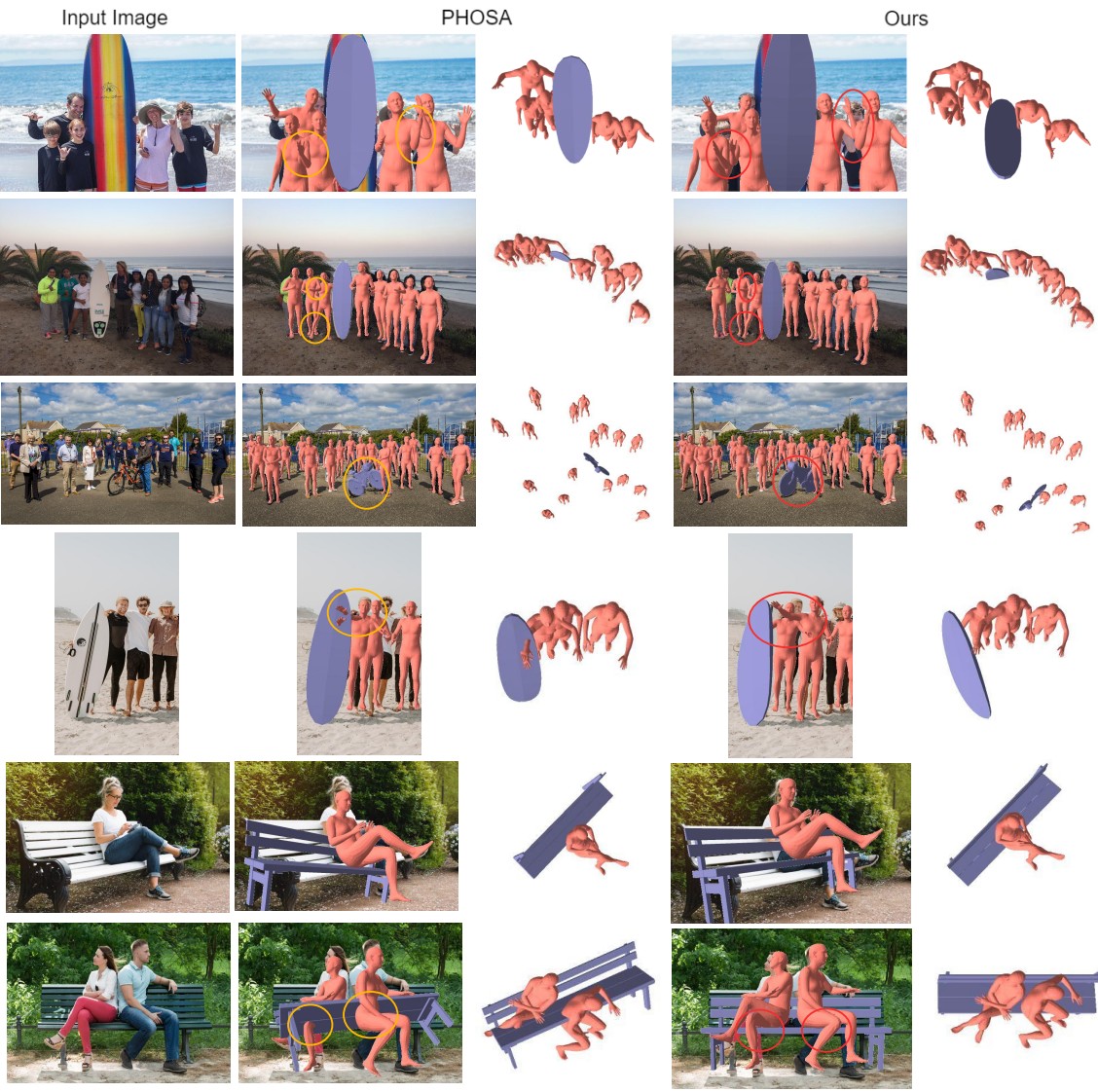

Figure 7: Our method, recovers plausible human-object and human-human spatial arrangements by explicitly reasoning about them. Here we demonstrate reconstruction on images with both humans and objects and compare PHOSA's reconstructions to those produced by our method.

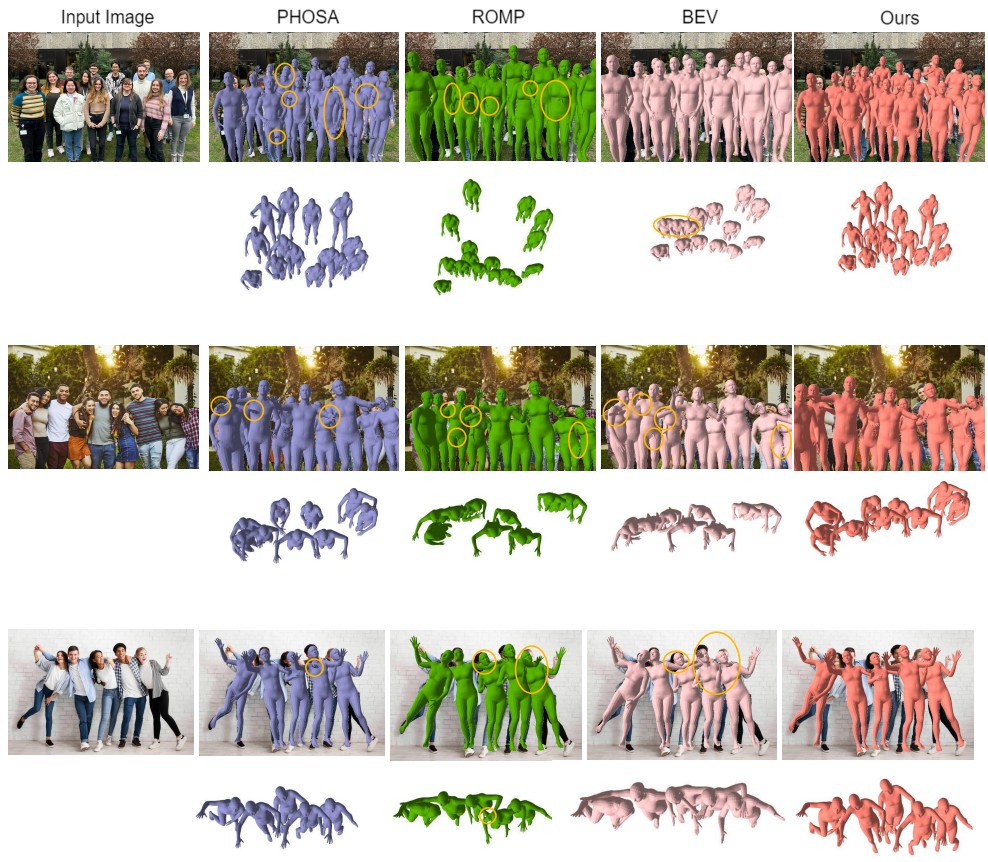

Figure 8: we illustrate the differences in human reconstructions generated by PHOSA, ROMP, BEV, and Our approach when provided with an input image. Our approach produces more plausible reconstructions with a substantial decrease in mesh collisions, all while maintaining relative coherence.

