# OpenReview forum: "SINGLE-IMAGE COHERENT RECONSTRUCTION OF OBJECTS AND HUMANS"
_ICLR.cc/2024/Conference — Submitted to ICLR 2024_

### Official Review · Reviewer_1ipB · 2023-10-31

**Soundness:** 2 fair
**Presentation:** 4 excellent
**Contribution:** 2 fair
**Rating:** 5
**Confidence:** 3

**Summary:**

This paper prsents a method to reconstruct 3D shape from a single 2D RGB image without explicit 3D supervision. This research mainly focuses on the scenes where multiple human and objects appear and they have interactions including human-human and/or human-objects. In the proposed pipeline, human and objects are reconstructed separately and later their relative arrangement is controlled through the joint optimization process. Several loss functions are introduced (collision loss, interaction loss, depth-ordering loss, occlusion-silhouette loss). The results are compared with previous reearches using COCO 2017 dataset.

**Strengths:**

* Well defined the loss functions essential to be considered to resonctruct 3D meshes from the single 2D image that contains interacting human and objects.
* The explanation was enough to understand the main idea and the intention. Easy to read and well prior works are well introduced.
* Using image inpainting algorithm to solve the problem of occluded object pose matching seems to be a clever idea and working well.

**Weaknesses:**

* Many of the building blocks of the proposed pipeline rely on the previously proposed researches. The authors claim that the novelty of collision loss, but since the collision loss is applied only after the reconstructions of each elements (human, object) are processed by the previously proposed method, the arrangement of the final scene seems to be too sparse to avoid collision between elements. The only arrangement can be optimized.
* I may assume another reason of sparse arrangement is the collision loss exploits compact 3D bounding box to reduce computational burden.
* One may think the quantitative evaluation metric is unfair in terms of comparison, since the metrics are designed to obtain high score according to the loss design.

**Questions:**

I think just obtimizing arrangement is not enough to get more reasonable results. Is there any method or idea to change the pose of each human considering the interaction instead of using each of 3D reconstruction meshes as pre-processed building blocks?

---

### Official Review · Reviewer_NYAH · 2023-10-31

**Soundness:** 3 good
**Presentation:** 3 good
**Contribution:** 2 fair
**Rating:** 3
**Confidence:** 2

**Summary:**

This paper presents a new pipeline for human object reconstruction from single-view images. This method mainly focuses on addressing occlusions, e.g., human-human interactions and human-object interactions, to produce a reasonable spatial layout. This method does not need any 3D supervision. Occlusion issues are addressed in an interaction optimization way. Many strategies are employed to estimate a correct human and objects, e.g., depth ordering, and image impainting.

From my side, this paper is a systematic work towards how to improve the spatial human and object layout placement with minor occlusion issues. Most technical ideas exist, which makes this paper a system and engineering work. The results look good. However, there lacks of comparisons with recent closely related works.

The paper is well writting.

**Strengths:**

This paper focuses on addressing occlusions for better human and object pose estimation. The major strength mainly comes from the human-object and human-human occlusion losses. However, many losses are inspired by the baseline [1]. Another strength is the mask inpainting for better 6 DOF pose estimation.

[1] Coherent reconstruction of multiple humans from a single image, CVPR2022

**Weaknesses:**

Even though this paper achieves convincing visualizations, there are quite a lot of weaknesses.

1. The technical contributions are not enough
The authors claim that they can address occlusion issues in many circumstances, e.g., human-human and human-object occlusion. However, the major methodology (loss function)  to achieve their target is heavily inspired/designed based on existing works, e.g. interaction loss and depth-ordering loss from [1].

2. Overclaimed contributions
For the first contribution point, I do not think it is the first approach. Please check [2, 3].
For the second contribution point, from my view, many loss items are borrowed from [1]. I can not see novelty there.

3. No discussions and comparisons with closely related papers.
The authors did not compare their performance with the recent works. There are quite a lot of works in human-object reconstruction. Since this is a fast-growing area, comparing it with a paper published in 2020 will not convince me. I would encourage the authors to compare with the state-of-the-art, e.g. [2,3]

[1] Coherent reconstruction of multiple humans from a single image, CVPR2022

[2] Human-Aware Object Placement for Visual Environment Reconstruction, CVPR2022

[3] Holistic 3D Human and Scene Mesh Estimation from Single View Images, CVPR2021

**Questions:**

Suggestions:
1. More comparisons with the state-of-the-art.
2. Rephrase the contribution claims.

---

### Official Review · Reviewer_HiJk · 2023-11-01

**Soundness:** 3 good
**Presentation:** 2 fair
**Contribution:** 2 fair
**Rating:** 5
**Confidence:** 4

**Summary:**

This paper presents a method for reconstructing a 3D scene containing multiple objects and humans. Compared to prior work such as [Jiang et al. 2020, Zhang et al. 2020], the main novelty is fitting a silhouette of a 3D mesh on inpainted instance segmentation, to improve accuracy on occluded objects. A holistic reconstruction through joint optimization is performed; consequently, both human-human and human-object occlusions and interactions can be considered.

**Strengths:**

It can leverage most off-the-shelf inpainting and segmentation models and does not require extra data or 3D supervision.

Well motivated. Reasoning about 3D geometry and affordances can improve most existing methods; it can also be potentially extended to generate targets for reinforcement or unsupervised learning.

**Weaknesses:**

Not enough experimental validation. It is possible I misunderstood, but there seems to be no comparison with more reasonable baselines like silhouette fitting without inpainting, or another human-human interaction method [Jiang et al. 2020].

While the statement that the inpainting approach "greatly boosts the precision of 6 DOF object position estimations" may not technically be a misrepresentation, I believe it is confusing. The actual evaluations show mesh collision metrics and user preference study.

**Questions:**

How important is inpainting performance? Why did you choose EdgeConnect for inpainting?

---

### Official Review · Reviewer_6q9o · 2023-11-06

**Soundness:** 2 fair
**Presentation:** 3 good
**Contribution:** 2 fair
**Rating:** 3
**Confidence:** 5

**Summary:**

In this paper is proposed a novel technique to infer the 3D reconstruction of multiple objects (this is a combination of humans and additional everyday objects) from a single image. The method first exploits two approaches to infer the 3D reconstruction of humans and objects, independently. After that, the paper presents an approach to resolve the ambiguities that could arise from the previous independent composition by means of a collision loss, depth ordering and interaction information.

**Strengths:**

+ The method handles an important problem in computer vision.  To this end, just a single image is used and no 3D supervision is considered. In contrast, a novel collision loss is exploited.
+ The segmentation mask is improved by a well-known inpainting-based approach. Thanks to that, the precision of 6 d.o.f object position in heavily occluded scenes is better.

**Weaknesses:**

- The full method seems to be a good combination of well-known approaches in the literature. In this line, I feel the authors should explain better their real contribution with respect to state of the art. Right now, the contribution seems to be minor, according to the information in the document.
- Some important analysis and experiments are missing in the paper.
- Some claims are not properly validated. The quantitative analysis shows that some of them were not solved as proposed.

**Questions:**

I think some of the claims in the paper are not properly validated. For instance, the authors comment, their method can handle strong occlusions and extreme interaction between humans and the relation human-object. Unfortunately, this interaction seems not to be correct as it can be seen in figure 2 (see hand interaction between people #1 and #2, from left to right). Maybe most important is the interaction between the first human (again from left to right) and the bicycle. In fact, no connection is inferred between human and bicycle in the provided solution. The same can be seen between the third human and the object. This analysis is very similar after observing the results in Fig. 5. On balance, the theoretical spatial coherence is only good in 2D, instead of 3D as the authors claim in the paper. In spite of improving the incoherent solutions of some competitors, I think the claims of the paper are not evaluated properly.

The authors infer a scale parameter per object (assuming, a human body is also an object), but to sort out the problem with multiple ones, I think a unique scale per scene (and camera) should be estimated. I feel it is the unique way to guarantee a coherence scale per scene.

Equation (1) is optimized just with respect to translation and scale. What about rotation? In my opinion, this is a key factor in the interaction between objects.

Can the authors explain their contribution in the collision loss? The final loss can be an extension of the work proposed by Jiang et al. 2020.

Robustness. How can the method handle bad detections and segmentations? Note that a claim of the paper is the use of complex scenes in the wild where potentially the detection and segmentation algorithms could fail, or at least, provide non-perfect estimations. How are these artifacts considered in the optimization?

According to experimental results, just three external types of objects are considered in the evaluation. In my opinion, the set of interactions needs to be more extensive, by using more objects and complex interactions.

Some failure cases could help readers.

---

### Meta-Review · Area_Chair_UdXX · 2023-12-07

**Metareview:**

This work tackled the task of jointly reconstructing humans and objects. The approach proposed is to first perform independent reconstruction via off-the-shelf systems and the refine via occlusion objectives. The reviewers raised several concerns regarding the contributions (as several prior works also use similar occlusion objectives) and lack of comparisons to recent related work, but the authors did not respond to address these. The reviewers all recommend rejection and the AC concurs.

**Justification For Why Not Higher Score:**

The reviews raised several concerns about the contributions and missing comparisons. Given that these were unaddressed, it is not possible to recommend acceptance.

**Justification For Why Not Lower Score:**

N/A

---

### Decision · Program_Chairs · 2024-01-16

Reject